# Long Island Sound Temperature Variability and its Associations with the Ridge-trough Dipole and Tropical Modes of Sea Surface Temperature Variability

Justin A. Schulte[1], Sukyoung Lee[2]

[1]Science Systems and Applications, Inc., Lanham, Maryland, 20706, United States
[2]Department of Meteorology, The Pennsylvania State University, University Park, 16802, United States

*Correspondence to*: Justin A. Schulte (justin.a.schulte@nasa.gov)

**Abstract.** Possible mechanisms behind the longevity of intense Long Island Sound (LIS) water temperature events are examined using an event-based approach. By decomposing an LIS surface water temperature
time series into negative and positive events, it is revealed that the most intense LIS water temperature event in the 1979-2013 period occurred around 2012, coinciding with the 2012 ocean heat wave across the mid-Atlantic Bight. The LIS events are related to a ridge-trough dipole pattern whose strength and evolution can be determined using a dipole index. The dipole index was shown to be strongly correlated with LIS water temperature anomalies, explaining close to 64% of cool-season LIS water temperature
variability. Consistently, a major dipole pattern event coincided with the intense 2012 LIS warm event. A composite analysis revealed that long-lived intense LIS water temperature events are associated with tropical sea surface temperature (SST) patterns. The onset and mature phases of LIS cold events were shown to coincide with central Pacific El Niño events, whereas the termination of LIS cold events was shown to coincide possibly with canonical El Niño events or El Niño events that are a mixture of eastern
and central Pacific El Niño flavors. The mature phase of LIS warm events was shown to be associated with negative SST anomalies across the central equatorial Pacific, though the results were not found to be robust. The dipole pattern was also shown to be related to tropical SST patterns and fluctuations in central Pacific SST anomalies were shown to evolve coherently with the dipole pattern and the strongly related East Pacific/North Pacific pattern on decadal time scales. The results from this study have
important implications for seasonal and decadal prediction of the LIS thermal system.

**1 Introduction**

Fluctuations in sea-surface temperature (SST) across coastal portions of the United States (U.S.) are driven by changes in oceanic and atmospheric circulation patterns. Changes in water temperature along the U.S. west coast are related to the Pacific Decadal Oscillation (PDO) and the North Pacific Gyre Oscillation, as well-documented (Mantua et al., 1997, Mantua and Hare, 2002, Di Lorenzo, 2008). For the U.S east coast, water temperature fluctuations are related to changes in the Gulf Stream position and variations in the Atlantic Multidecadal Oscillation, PDO, and East Pacific/North Pacific (EP/NP) pattern (Pershing et al., 2015, Schulte et al., 2017). Superimposed on the water temperature changes driven by natural modes of variability is background warming associated with anthropogenic climate change (Pershing et al., 2015).

Although the mechanisms behind SST variability along the U.S. west coast are well-documented, comparatively fewer studies have focused on understanding SST variability across the mid-Atlantic Bight in the context of large-scale climate modes. Two recent studies put water temperature variability across the Gulf of Maine and the Long Island Sound (LIS) in a climate-mode context. The first study by Pershing et al. (2015) showed that the combination of Gulf Stream and PDO influences led to rapid warming of the Gulf of Maine that resulted in the collapse of the cod fishery.

More recently, a second study by Schulte et al. (2018) found the EP/NP pattern to be a dominant pattern governing LIS water temperature variability. The EP/NP pattern was shown to be strongly correlated to LIS water temperature unlike the well-known North Atlantic Oscillation (NAO; Hurrell, 1995), Pacific North American (PNA; Wallace and Gutzler, 1981, Svoma, 2011), Arctic Oscillation (AO; Thompson and Wallace, 1998), and West Pacific (WP; Barston and Livezey, 1987; Linkin and Nigam, 2008) patterns. In fact, Schulte and Lee (2017) found that the EP/NP pattern is more strongly related to temperature variability across the Northeast U.S than the AO, which is often associated with colder-than-normal conditions across the region (Wettstein and Mearns, 2002). Those results suggest that the EP/NP pattern is an important component to seasonal prediction of air and water temperature across the LIS region. Another important aspect of the EP/NP pattern is its strong decadal variability, which could enable decadal prediction of LIS water temperature. Schulte et al. (2018) termed the decadal component of the EP/NP pattern the quasi-decadal mode and showed that it fluctuates coherently with LIS water

temperature anomalies. The physical mechanism contributing to the EP/NP decadal variability was not identified, underscoring the need for an additional study to identify a possible source of the EP/NP decadal variability. Understanding the mechanisms behind the EP/NP decadal variability has implications for seasonal and decadal prediction of LIS water temperature.

5       Improving the current understanding of the climatic mechanisms governing LIS water temperature variability also has important implications for managing fisheries. For example, Howell and Auster (2012), using finfish abundance indices, found shifts in spring community structures that are related to water temperature. The LIS American lobster, which is sensitive to water temperature, dramatically declined around 1997 (Pearce and Balcom, 2005), but managing the lobster harvests has failed to recover

10   the lobster fishery. For the nearby Rhode Island Sound, biological communities are related to spring-summer water temperature (Collie et al. 2008), suggesting that predicting spring-summer water temperature could aid the setting of fish harvest quotas. These studies underscore the need to better understand water temperature variability across the LIS region so that changes in biological communities can be better monitored and used to better manage fish harvests.

15       Another reason that the LIS is important to study is that it is in a region where air temperature and precipitation are not strongly influenced by well-known climate modes such as the NAO, AO, PNA, and WP that are extracted from the widely used classical empirical orthogonal function analysis (EOF) method (Barston and Livezey, 1987). Schulte et al. (2016) found weak relationships between well-known climate indices and variability of precipitation and temperature across the Northeast U.S. As shown by

20   Schulte et al. (2018), LIS water temperature variability is strongly related to neither changes in the Gulf Stream position nor fluctuations in the NAO despite being located adjacent to the Atlantic Ocean. The weak Gulf Stream influence is likely the result of the LIS being a semi-enclosed water basin, whereas the general movement of weather systems from west to east may reflect the weak NAO influences because the NAO's centers of action are located downstream of the LIS.

25       Recognizing that well-known climate indices are weakly related to the salinity variability of Northeast U.S. estuaries, Schulte et al. (2017a) adopted a continuum approach (Franzke and Feldstein, 2005; Johnson and Feldstein, 2010; Johnson et al., 2008) to teleconnection pattern extraction and identified an Eastern North American Sea Level Dipole (ENA) pattern that is more strongly correlated

with streamflow than the PNA and NAO indices. In a subsequent study, Schulte et al. (2017b) found the ENA pattern to be strongly related to Northeast U.S. precipitation and LIS salinity. Those results suggest that a continuum approach is better suited for understanding climate variability and associated LIS water temperature impacts than an EOF-based analysis. Although Schulte et al. (2018) did show that the EOF-based EP/NP pattern is strongly correlated with LIS water temperature, the EP/NP pattern for December cannot be unambiguously extracted using the Rotated Principal Component analysis (RPCA) conducted by the Climate Prediction Center (CPC). Furthermore, EOF analysis assumes atmospheric patterns are orthogonal even though orthogonality does not hold for the real atmosphere. This orthogonality assumption can lead to the generation of unphysical modes (Tremblay, 2001). In contrast, clustering methods such as Self-Organizing Maps (SOMS) that view the atmospheric as a continuum more accurately produce patterns that are actually observed than EOF analysis (Liu et al., 2006; Johnson et al., 2008, Yuan et al., 2015).Therefore, an additional study is needed to construct an atmospheric circulation index that is strongly related to LIS water temperature variability, physically based, and unambiguously defined for all months.

In this paper, we use an event-based approach to identify LIS water temperature relationships with atmospheric and oceanic patterns. More specifically, the main objectives of the study are the following: (1) identify the atmospheric circulation patterns associated with LIS water temperature events; (2) create a simple atmospheric index that is strongly correlated with LIS water temperature variability; and (3) use the simple atmospheric index to better understand LIS water temperature variability. Because tropical Pacific SST patterns are often used in seasonal forecasting over North America, in this study we also explore if there is a SST-pattern precursor to LIS temperature events.

## 2 Data

In this paper, SST fields from 1870 to 2013 are based on the Hadley Centre Global Sea Ice and Sea Surface Temperature (HadISST1) data set (Rayner et al., 2016). Atmospheric fields were analyzed using 500-hPa geopotential height and sea-level pressure (SLP) fields based on the National Oceanic Atmospheric Administration's 20[th] century Reanalysis (Compo et al., 2011) and the National Center for Atmospheric Prediction (NCEP; Kalnay et al., 1996) Reanalysis. The 20[th] century Reanalysis product

was used because the product extends back to 1851, whereas the NCEP Reanalysis product only extends back to 1948. Mean monthly air temperature data from 1979 to 2013 were based on the observed U.S climate divisional data set (Guttman and Quayle, 1996). The data set comprises average monthly temperature data for 344 climate regions (Figure 1a) that partition the U.S into homogeneous climate zones. The annual cycles were removed from the data by subtracting the mean monthly values for each month from the monthly values of the corresponding month for each grid point or climate division.

LIS surface water temperature data used in this study were generated from the New York Harbor Observing Prediction System (NYOPS; Georgas et al., 2016) model. Model generated data were preferred to observational data in this study because observations are temporally sparse and continuous data are needed for the methods adopted in this study. The NYHOPS model is a three-dimensional hydrodynamic model with 11 vertical levels. Following Schulte et al. (2018), water temperature computed on the 1$^{st}$ vertical level was considered surface water temperature. The LIS is a well-mixed estuary, so the choice of vertical level is not critical to the results presented in this study. To obtain a single time series representing LIS surface water temperature (for brevity, referred to as LIS temperature, hereafter), water temperature was averaged over the gray-shaded region shown in Figure 1b. The annual cycle in the resulting LIS temperature time series was removed using 1979-2013 monthly means.

The LIS temperature time series and the SST fields were detrended to remove the long-term trend. The time series were detrended by fitting a least-square fit of a line to the time series and subtracting the line from the time series. To check the sensitivity of results to detrending, the analyses were conducted using both the detrended and non-detrended data. Results for the detrended analysis are shown unless otherwise specified. The reason for showing the detrended results is that the study is focused on interannual variability rather than long-term trends.

Indices for the NAO, AO, EP/NP, PNA and the WP were obtained from the CPC and were based on the 1979-2013 period. The NAO, WP, PNA, and EP/NP indices obtained from the CPC were calculated from a RPCA of 500-hPa geopotential height anomalies poleward of 20°N. The AO index was calculated from a rotated RPCA analysis of 1000-hPa geopotential height anomalies. Data for the 1950-2013 period were also used for the EP/NP index.

The Niño 3 and Niño 4 indices from 1870 to 2013 (available at https://www.esrl.noaa.gov/psd/gcos_wgsp/Timeseries/) were used to measure the strength and evolution of the El-Niño/Southern Oscillation (ENSO). Whereas the Niño 3 index better describes the evolution of canonical ENSO, the Niño 4 index better describes the evolution of central Pacific ENSO events (Kao and Yu, 2009; Lee and McPhaden, 2010) Thus, using these two indices, we accounted for two different flavors of ENSO. The annual cycles from these ENSO metrics were removed using the long-term (1870-2013) monthly means.

## 3 Methods

### 3.1 Event Decomposition

To better understand the characteristics of climate time series, time series were decomposed into negative and positive events. More specifically, let a time series $X$ be a sequence of $N$ data points $x_1, x_2, \ldots, x_N$ at the time points $t_1, t_2, \ldots, t_N$, where the data points were assumed to be equally spaced. Data points were based on monthly anomalies so that they were both positively and negatively valued. Thus, a complete sequence $x_1, x_2, \ldots, x_N$ was partitioned into subsequences comprising adjacent data points whose values are of similar sign. Such subsequences were termed positive or negative events depending on the values of the data points.

The onset and decay of events were defined as follows. A negative event $E_{neg}$ was said to begin at $t_j$ if $x_j < 0$ and $x_{j-1} > 0$. A negative event beginning at $t_j$ was said to terminate at $t_k \geq t_j$ if $x_k < 0$, $x_{k+1} > 0$, and $x_i < 0$ for all $i$ such that $j \leq i \leq k$. A similar definition was used to define positive events, but the sign conventions were reversed. The time point $t_j$ was termed the onset phase and the time point $t_k$ was termed the decay phase. The peak intensity of a negative (positive) event was deemed the minimum (maximum) value obtained by a data point within the event period $[t_j \ t_k]$. If the peak intensity of an event occurred at $t_p$, then $t_p$ was termed the mature phase.

Given this definition of an event, an event occurring over the time period $[t_j \ t_k]$ contained $M = t_k - t_j + 1$ data points, where the integer $M$ was regarded as the persistence of the event. The cumulative intensity (referred to as the intensity, hereafter) of an event $E$ was defined as

$$I = \sum_{i=1}^{M} y_i, \tag{1}$$

where the $y_i$ are data points composing the event $E$. The absolute value of intensity was deemed the magnitude of an event. The duration and intensity of events were depicted using an event spectrum. The event spectrum was comprised of line segments beginning at the onset phases and ending at the termination phases of the events. That is, for each event, a line segment was drawn from the point $(t_j, I)$ to the point $(t_k, I)$ so that the length of the line segment represented the event duration.

There are several advantages to using the event decomposition approach. The first advantage is that the autocorrelation of the data is accounted for by grouping the data into events. The second advantage is that persistence of individual events can be readily defined, whereas the lag-1 autocorrelation coefficient measure of persistence needs to be calculated using a data interval so that the lag-1 autocorrelation coefficient may not reflect the persistence of an individual event. A third advantage is that potential nonlinearities are accounted for by analyzing negative and positive events separately.

**3.2 Wavelet Analysis**

To extract time-frequency information from a time series $X$, a wavelet analysis (Torrence and Compo, 1998) was implemented. The wavelet transform of $X$ is given by

$$W_n^X(s) = \sqrt{\frac{2\delta t}{s}} \sum_{n'=1}^{N} x_{n'} \, \psi^* \left[ \frac{(n'-n)\delta t}{s} \right], \tag{2}$$

where $\psi$ is the Morlet wavelet given by

$$\psi(\eta) = \pi^{-1/4} e^{i\omega_0 \eta} e^{-\frac{1}{2}\eta^2}, \tag{3}$$

$\omega_0 = 6$ is the dimensionless frequency, $t$ is time, $s$ is wavelet scale, $\delta t$ is a time step (1 month in this study), $\eta = s \cdot t$, and the asterisk denotes the complex conjugate (Torrence and Compo, 1998).

To quantify the relationships between climate modes and water temperature as a function of frequency and time, a wavelet coherence analysis was conducted. Following Grinsted et al. (2004), the (local) wavelet squared coherence between two time series $X$ and $Y$ is given by

$$R_n^2(s) = \frac{\left| S(s^{-1} W_n^{XY}(s)) \right|^2}{S(s^{-1} |W_n^X(s)|^2) \cdot S(s^{-1} |W_n^Y(s)|^2)}, \tag{4}$$

where $W_n^{XY}(s)$ is the cross-wavelet transform defined as the product of the wavelet transform of $X$ and the complex conjugate of the wavelet transform of $Y$. In Eq. (4), $S$ is a smoothing operator that smooths

coherence in time and in wavelet scale (Grinsted et al., 2004). Using Monte Carlo methods, the statistical significance of wavelet squared coherence was assessed by generating 10000 pairs of surrogate red-noise time series with the same lag-1 autocorrelation coefficients as the input time series and computing the wavelet squared coherence between each pair (Grinsted et al., 2004).

5   To reduce the number of false positive results arising from the simultaneous testing of multiple hypotheses (Maraun and Kurths, 2004; Maraun et al., 2007, Schulte et al., 2015; Schulte, 2016), the cumulative area-wise test developed by Schulte (2016) was applied. The test tracked how the areas of contiguous regions of pointwise significance (significance patches) changed as the pointwise significance level was altered. The test was applied by computing the normalized areas of pointwise significance

10 patches over a discrete set of pointwise significance levels. The normalized area of a patch was defined as the patch area divided by the square of its centroid's scale coordinate (Schulte et al., 2015; Schulte. 2016). In this study, the normalized areas were computed using pointwise significance levels ranging from $\alpha = 0.02$ to $\alpha = 0.18$. The spacing between adjacent pointwise significance levels was set to 0.02.

   The strength of coherence was also measured using global coherence (Schulte et al., 2016), the

15 time-averaged representation of local wavelet squared coherence. Global coherence is given by

$$G_C(s) = \frac{|W^{XY}(s)|^2}{\left(\sum_{n=1}^{N}|W_n^X(s)|^2\right)\left(\sum_{n=1}^{N}|W_n^Y(s)|^2\right)}, \tag{6}$$

where

$$W^{XY}(s) = \sum_{n=1}^{N} W_n^X(s)W_n^{Y*}(s). \tag{7}$$

(Schulte et al., 2016). The statistical pointwise significance of $G_C(s)$ was computed using Monte Carlo

20 methods in a similar manner to local wavelet squared coherence.

   A lower dimensional version of the cumulative area-wise test was applied to the global coherence spectra to reduce the number of false positive results (Schulte et al., 2018). The test assessed the statistical significance of one-dimensional arcs using arc length, which is an integrated metric accounting for the width of the peak in wavelet scale (frequency) and the extent to which the peak is above the critical level

25 of the pointwise test. To track how the arc length of a given pointwise significance peak changed as the pointwise significance level was altered, the arc length of the pointwise significance peak was computed at pointwise significance levels ranging from 0.02 to 0.18. The test statistic in this case is cumulative arc

length. Normalized arc lengths were used to account for how peaks widen with wavelet scale. This normalization was achieved by computing the logarithm (base 2) of the wavelet scales. To further normalize, global coherence values at each wavelet scale were divided by the critical level of the test associated with the pointwise significance level 0.02 at each wavelet scale. The null distribution for the cumulative arcwise test was obtained by generating surrogate red-noise timeseries in the same manner as the cumulative area-wise test. For reference, we also plotted the traditional 5% pointwise significance bounds on all global spectra plots in this study. The reader is referred to Schulte et al. (2018) for more details regarding the cumulative arcwise test.

## 4 Results

### 4.1 LIS Temperature Time Series

The time series of detrended LIS temperature anomalies is shown in Figure 2a. Some notable features are the cool periods around 1982,1996, 2003 (thick blue lines) and the warm events around 1991, 2001, and 2012. The 1982 cool period is rather intense, with LIS temperature anomalies approaching -2°C. In contrast, the 2012 event is associated with a maximum temperature anomaly of approximately 3°C, making this water temperature anomaly the largest in the 1979-2013 period.

Unlike the simple time series shown in Figure 2a, the event spectrum shown in Figure 2b clearly distinguishes the intense LIS events from the weak short-lived events. For example, both the cool period around 1982 and the warm episode around 2012 emerge as the most intense cool and warm events in the 1979-2013 period. The 1996, 2003, and 2011 cold events are nearly as intense as the 1982 event (Table 1). It is interesting to note that the most intense negative events shown in Table 1 peak in winter (December-February). This tendency for negative events to peak in winter was confirmed by computing the number of times a negative event peaked in a given month for a larger set of events (32 events) whose intensities fall below the median of all negative event intensities. A similar but weaker tendency was found for positive events, with a majority of the most intense (greater than 50[th] percentile) positive events peaking in January and February.

The event spectrum also allows for the clear comparison of event persistence. An inspection of Figure 2b shows that the intense events are generally more persistent than less intense events. In fact,

Table 1 shows that all the most intense events have persistence of at least 5 months, exceeding the average persistence of 3 months calculated using all events. Negative and positive events were found to have similar average persistence. Given that the average persistence is 3 months, the 1982, 1991, 1996, 2003, and 2012 LIS temperature events are unusually persistent compared to other events in the study period. Moreover, not only is the 2012 event among the most persistent, but also its magnitude far exceeds that of any other event even with the removal of the long-term trend. This result suggests that atmospheric variability may have contributed strongly to that event.

**4.2 LIS Events and Atmospheric Patterns**

To diagnose a possible mechanism behind the occurrence of the intense LIS events, the correlation between 500-hPa geopotential height anomalies and detrended LIS temperature anomalies was computed. We used this one-point correlation approach to extract a relevant teleconnection pattern from a continuum of patterns, much like how the PNA pattern was originally derived (Wallace and Gutzler, 1981). For this analysis, we focused on the December-February (DJF) season because atmospheric circulation anomalies are generally most pronounced during the DJF season and because the peak of events tends to occur in the winter.

Shown in Figure 3a is the correlation between DJF LIS temperature anomalies and 500-hPa geopotential height anomalies. The positive correlation between LIS temperature anomalies and 500-hPa geopotential height anomalies over the eastern U.S. suggest that warmer-than-normal LIS temperature conditions are associated with a jet stream ridge, which is consistent with how jet stream configurations can influence the temperature in the coastal ocean off the northeastern U.S. (Chen et al., 2014). Similarly, the negative correlations with 500-hPa geopotential height anomalies across Alaska indicate that LIS warm events are associated with an anomalous trough over Alaska. Thus, it appears that LIS temperature events are related to a ridge-trough dipole pattern, an anomalously amplified wave pattern across the U.S.

For comparison, the 500-hPa geopotential height anomaly field for the March 2012 event is shown in Figure 4a. Negative 500-hPa geopotential height anomalies are seen across Alaska, and positive 500-hPa geopotential height anomalies are seen across the eastern U.S. and across the central North Pacific. This 500-hPa geopotential height anomaly configuration is consistent with the results presented in Figure

3a, suggesting that the ridge-trough dipole pattern was an important contributor to the March 2012 LIS temperature event.

A physical mechanism behind the ridge-trough dipole-LIS temperature association can be better diagnosed by examining the relationship between DJF SLP anomalies and LIS temperature anomalies. As shown in Figure 3b, DJF LIS temperature anomalies are negatively correlated with DJF SLP anomalies across north western Canada. Thus, positive LIS temperature anomalies are associated with anomalous cyclonic flow, whereas negative LIS temperature anomalies are associated with anomalous anticyclonic flow. These relationships are physically consistent with findings from previous works showing how upstream (relative to the LIS) surface anti-cyclones play a crucial role in the occurrence of cold temperature extremes across the Northeast U.S. (Konrad, 1996, Konrad, 1998). Such surface anti-cyclones have been shown to be dynamically supported by a ridge-trough wave pattern in the middle troposphere (Konrad, 1998; Jones and Cohen 2011), in agreement with the results shown in Figure 3a. At the surface, the anti-cyclone results in cold advection from a high latitude source region into the LIS region (Figure 5a). Cyclones are also typically present along the East Coast U.S. during cold temperature events (Konrad, 1998), which could explain the positive correlation between LIS temperature anomalies and SLP anomalies along the East Coast U.S. Features associated with positive LIS temperature anomalies are opposite to those found for negative LIS temperature anomalies (Figure 5b).

As a specific example, consider the SLP anomaly pattern for March 2012 (Figure 4b). Negative SLP anomalies are seen to extend from western Alaska to central Canada, indicative of anomalously strong cyclonic flow and warm air advection across the eastern U.S. The location of the negative SLP anomalies generally coincide where SLP anomalies are correlated to LIS temperature anomalies (Figure 3b), again suggesting that the ridge-trough pattern played an important role in the March 2012 LIS temperature event.

To better understand LIS water temperature variability, a ridge-trough dipole index was created based on the pattern identified in Figure 3a. The dipole index was constructed by first locating the grid point for which the correlation between LIS temperature and 500-hPa geopotential height anomalies is minimum. This grid point is located at 70°N and 157.5°W and is marked by a cyan cross in Figure 3a. Next, the grid point for which the correlation between LIS temperature and 500-hPa geopotential height

anomalies is maximum was located. This grid point is located at 42.5°N and 75°W and is marked with a magenta cross in Figure 3a. Following Wang et al. (2014), the dipole index for a given month was defined as the 500-hPa geopotential height anomaly at 42.5°N and 75°W minus the 500-hPa geopotential height anomaly at 70°N and 157.5°W. Thus, the dipole index measures the intensity of the ridge-trough dipole

pattern such that positive phases generally indicate that an anomalously strong ridge over the eastern U.S is accompanied by an anomalous trough over Alaska. Correlating the dipole index with SLP anomalies (not shown) reveals that negative (positive) phases of the dipole pattern are associated with positive (negative) SLP anomalies across north western Canada and cold (warm) air advection to the east of the anomaly center (Figure 5).

10        The time series for the 3-month running mean of the dipole index is shown in Figure 6. The time series is rather noisy, but notable features can still be identified. The dipole pattern, as indicated by positive dipole indices, is seen to be in a persistent positive phase around the 2012 LIS warm event. The positive dipole event around 2012 is quite intense but not as intense as the 1882 positive dipole event that persists for 11 months (Table 2). The most intense negative dipole event occurs around 1977. A

comparison of Tables 1 and 2 shows that the second most intense negative dipole event calculated using the raw monthly dipole index time series coincides with the second most intense LIS cold event that occurred around 2003 (Table 1).

        Although a comparison of Tables 1 and 2 shows that the 2012 LIS warm event coincides with the third most intense positive dipole event, the relationship strength between LIS temperature anomalies and

the dipole pattern cannot be inferred. How strongly related is the dipole index to LIS water temperature anomalies? To assess the strength of the dipole index relationship with LIS water temperature anomalies, seasonally averaged detrended LIS temperature anomalies were correlated with the seasonally averaged dipole index. As shown in Figure 7, the dipole index is strongly correlated with LIS temperature anomalies for the October-December (OND), November-January (NDJ), DJF, and January-March (JFM)

seasons. The relationships are generally stronger if the dipole index leads by 1 month, as indicated by the dotted line in Figure 7. The lagged correlation coefficients approach 0.8 for the DJF season, suggesting that DJF LIS temperature anomalies are strongly influenced by the dipole pattern in the NDJ season. Lagged correlations are also strong ($r > 0.6$) for the OND, NDJ, and JFM seasons. The relationships are

generally weaker in the warm seasons possibly because teleconnection patterns are generally of the weakest amplitude in the warm season. Another reason is that LIS temperature anomalies in the winter can persist into the summer (Schulte et al., 2018), weakening the simultaneous relationships between LIS temperature anomalies and the dipole pattern for non-winter seasons. However, the weaker relationships between air temperature and the dipole index in the summer (not shown) likely contribute to the seasonal cycle in relationship strength shown in Figure 7.

The relationship strength found between the dipole index and LIS temperature anomalies is consistent with results found in prior work because the dipole index is strongly correlated with the EP/NP index for most months (Table 3) and because the EP/NP index is strongly correlated with LIS temperature anomalies (Schulte et al., 2018). However, Schulte et al. (2018) used an ad hoc approach to construct the December EP/NP index using linear regression. Thus, one may ask: is the EP/NP pattern a wintertime pattern? To answer the question, it is first noted that the relationship strength between EP/NP and dipole indices increases nearly monotonically from August to November, decreases almost monotonically from January to August, and peaks in January. With a such clear seasonal cycle, it is natural to infer that the December dipole pattern is largely the December EP/NP pattern despite how the CPC suggests that the EP/NP pattern is inactive in winter. Because the 500-hPa geopotential height anomaly structure associated with the dipole pattern is the same from November to March (not shown), one can deduce that December ridge-trough pattern is largely the December EP/NP pattern given the strong relationship between EP/NP and dipole indices during those months. More specifically, we compared the December dipole pattern to the January dipole pattern (Figure 8) and found them to be similar in terms of 500-hPa geopotential height anomalies. Thus, because the January dipole and EP/NP indices are strongly correlated, the December dipole pattern must also be EP/NP like. The strong relationships ($r > 0.8$) between air temperature and the dipole pattern in winter suggests that the dipole pattern (or EP/NP pattern) is particularly active in the winter. It also has a strong physical basis because negative phases are associated with 500-hPa western North America and Arctic ridging, and eastern U.S. 500-hPa troughing (Figure 8), all features associated with eastern U.S. cold temperature events (Konrad, 1998). Although Schulte et al. (2018) used the EP/NP index to diagnose historical LIS temperature variability, our dipole index is defined in all months and is easy to calculate, making it more practical in an operational forecasting setting.

While the December dipole index is well-correlated with December indices for the AO, NAO, and WP, a close examination of the 500-hPa geopotential height anomaly fields (not shown) associated with those patterns reveals that those patterns are quite different from the dipole pattern found in this study. For example, the NAO and AO indices are correlated 500-hPa geopotential height around the dipole pattern's eastern center of action (green cross in Figure 3a) but are practically uncorrelated with 500-hPa geopotential height anomalies around the western center of action. The NAO and AO are also much more strongly correlated with 500-hPa geopotential height anomalies across the North Atlantic than the dipole pattern. The WP index is only weakly correlated with 500-hPa geopotential height around both dipole pattern's centers of actions and much more strongly corelated with 500-hPa geopotential height anomalies across the western North Pacific Ocean. Our results suggest that the reason why the wintertime AO, NAO, and WP patterns are not strongly related with wintertime LIS temperature anomalies as shown by Schulte et al. (2018) is that these patterns are not related to northern Alaskan jet-stream ridging that is important to LIS temperature variability (Figure 3a).

The fact that the dipole index is correlated with multiple large-scale indices (Table 3) suggests that the dipole pattern falls on a continuum of teleconnection patterns (Franzke and Feldstein, 2005) such that the dipole pattern is strongly EP/NP-like. Although we found by conducting our own EOF analysis of 500-hPa geopotential height anomalies that the AO pattern is the leading mode of variability, the EP/NP pattern appears to be consistently the 5th to 7th leading mode of variability. Thus, although the EP/NP pattern is not as dominant as the AO pattern, the dipole pattern tends to more closely resemble it than the AO pattern. Furthermore, the continuum-based extraction of a dipole pattern with a strong relationship to LIS temperature anomalies supports the idea that the continuum approach is useful for understanding climate variability across the Northeast U.S., a finding like that found in previous work focusing on precipitation in the Northeast U.S. (Schulte et al. 2017a, b). In particular, our results show that the one-point correlation map approach used by Wallace and Gutzler (1981) is a powerful but simple tool for understanding regional climate variability.

Given that LIS water temperature is strongly correlated with air temperature (Schulte et al., 2018), it is hypothesized that the dipole index is related to air temperature across the U.S., especially around the LIS. To confirm a dipole index-air temperature relationship, the dipole index was correlated with average

monthly air temperature anomalies for the 1979-2013 period (Figure 9). The results for the NDJ season are only displayed because the strongest correlation found in Figure 7 is between the NDJ dipole index DJF LIS water temperature anomalies.

As shown in Figure 9, the dipole index is, indeed, strongly correlated with air temperature anomalies across a large region of the U.S. Correlation coefficients exceed 0.8 and approach 0.9 across the Northeast U.S. and LIS region. The strong relationships extend to the southern U.S., and the relationships generally weaken equatorward. The relationships displayed in Figure 9 are generally stronger than those associated with the AO and NAO (not shown) whose influence on eastern U.S temperature has been well-studied (Hurrell and van Loon 1997, Wettstein and Mearns, 2002). Thus, our dipole index is useful for diagnostic studies of cold outbreaks across the eastern U.S. The results for the other seasons are similar, but the relationships for seasons not comprising November, December, January, February (e.g. June-August) are generally weaker than those identified for the NDJ season. This result suggests that the dipole pattern is rather dominant in the winter. The strong relationship between the dipole index and U.S. air temperature anomalies is consistent with how the intense 2012 dipole event coincides with the record warm March 2012 (Dole et al., 2014), which resulted in a so-called false spring in which plants bloomed prematurely making them susceptible to drought and freezes (Ault, 2013). The results shown in Figure 9 suggest that the dipole pattern's impact on LIS temperature is related to the dipole pattern's influence on air temperature.

### 4.3 Intense LIS Events and SST Patterns

SST patterns are often used in seasonal forecasting and, thus, identifying an SST pattern precursor to LIS temperature events has implications for seasonal prediction of LIS temperature anomalies. To identify SST patterns associated with LIS temperature events, a lagged SST composite analysis was conducted using detrended LIS warm and cool events separately. The SST composite plots for the warm events were constructed using the LIS warm events whose intensities are greater than or equal to the 50[th] percentile of all warm event intensities (32 events). Similarly, the SST composite plots for the cold events were constructed using LIS cold events whose intensities are less than or equal to the 50[th] percentile of all LIS cold event intensities (32 events). The composite mean SST patterns were computed at the onset,

mature, and decay phases of the LIS events. Because intense LIS events tend to peak in winter, the composite plots for mature phases mainly reflect wintertime conditions.

The results for the LIS cold events are shown in Figure 10. The composite plot shown in Figure 10a indicates that the onset of LIS cold events is associated with positive SST anomalies across the central equatorial Pacific. The results suggest that LIS cold events could be initiated by central Pacific El Niño events (Lee and McPhaden, 2010). A few examples of central Pacific El Niño events (based on the December-March season) are the 1991–92, 1994–95, 2002–03, 2004–05, and 2009-2010 events (Table 5), but a more complete list can be found in Johnson and Kosaka (2016). The 2002-2003, 2004-2005, and 2009-2010 events all appear to occur around LIS cold periods (Figure 2a). Note that there could be lags between the onset of central Pacific El Niños and LIS temperature anomalies because of the lagged response of water temperature to atmospheric forcing (Schulte et al., 2018). In addition, pre-existing positive water temperature anomalies may need time to degrade.

The SST anomaly pattern for the mature phases of LIS cold events features positive SST anomalies across the central equatorial Pacific (Figure 10b). However, the mature-phase composite mean SST anomaly pattern is more pronounced across the North Pacific Ocean than it is for the onset phase. A region of positive SST anomalies is seen to be horseshoe-shaped, with positive SST anomalies extending from the central equatorial Pacific to the U.S. west coast. Although Hartmann (2015) found an SST pattern resembling that shown in Figure 9b to be a contributor to the February 2015 eastern U.S. cold event, only a single event was considered. In this study, we show that the pattern is associated with numerous LIS negative temperature events (and thus eastern U.S. air temperature events), many of which persist for more than 5 months. Thus, we show here that the SST pattern influences both the intensity and persistence of events. It is noted that the pattern shown in Figure 10b resembles the DJF SST pattern of 1996, which is consistent with how 1996 was a cooler-than-normal period for much of the U.S. (Haplert and Bell, 1997) and the LIS (Table 1).

Unlike the composite mean SST pattern corresponding to the onset phase, negative SST anomalies are present along the U.S. east coast and across the Gulf of Mexico during mature phases. These results are consistent with how LIS temperature anomalies are strongly associated with the dipole pattern that influences air temperature across regions adjacent to the Gulf of Mexico and U.S. east coast. This

relationship between the dipole index and SST anomalies was confirmed by correlating the dipole index with SST anomalies for different seasons (not shown).

The tropical SST pattern associated with the decay phase of LIS cold events is different from those associated with the onset and mature phases (Figure 10c). The composite mean SST anomaly pattern most closely resembles the first leading mode of SST variability called the canonical ENSO pattern (Hartmann, 2015), though the most intense positive SST anomalies are still confined to the central equatorial Pacific. This result suggests that there may be a tendency for the decay of LIS cold events to coincide with canonical ENSO patterns or an SST pattern that is a mixture of central and eastern Pacific El Niño flavors lying on a continuum of ENSO flavors (Johnson, 2013).

The tendency for the decay of LIS cold events to coincide with canonical ENSO patterns is more evident when constructing SST composites using the $10^{th}$ percentile (Figure 11) instead of the $50^{th}$ percentile used to construct the composite shown in Figure 10. However, possibly because of small sample sizes (7 events), the results generally lack statistical significance. Nonetheless, the event spectrum depicted in Figure 2b indicates that the major cool period around 1982 and 1997, for example, terminate around the major 1982/83 (Ramusson and Wallace, 1983; Quiroz, 1983) and 1997/1998 (McPhaden, 1999) El Niño events. Although Schulte et al. (2018) showed that LIS temperature anomalies are associated with a single SST pattern, we show in this study that predicting the evolution of LIS temperature events may require knowledge of several ENSO flavors.

The SST pattern across the Atlantic Ocean shown in Figure 11 resembles a well-documented North Atlantic tripole mode (Deser and Blackmon, 1993, Fan and Schneider, 2011), which comprises three anomaly centers, one located off the south eastern U.S coast, a second one located east of Newfoundland, and a third one located in the tropical east Atlantic. This tripolar SST mode has been shown to be related to ENSO, the NAO, and local wind forcing (Fan and Schneider, 2011). The association between the tripole pattern and the LIS water temperature could reflect weak influences of the NAO on LIS water temperature. This interpretation is consistent with how the NAO and dipole patterns are related in the winter (Table 3). However, these Atlantic SST anomalies generally lack statistical significance, and this finding is consistent with how the LIS water temperature anomalies are only weakly related to the NAO.

The composite analysis was also conducted for LIS warm events, and the results revealed that LIS warm events are also associated with SST modes of variability (Figure 12). The onset of LIS warm events appears not to be associated with any coherent SST pattern. For the mature phase, statistically significant negative SST anomalies are seen across the central equatorial Pacific and positive SST anomalies are seen across the eastern equatorial Pacific. Like the SST anomaly pattern associated with mature phases of LIS cold events (Figures 10b and 11b), the pattern shown in Figure 12b generally resembles the 3[rd] leading mode of SST variability (Hartman, 2015). The SST pattern corresponds well with the SST anomaly pattern associated with March 2012 (Figure 4c), a month in which record warmth was experienced across the central and eastern U.S. (Dole et al., 2014). Mature phases are also associated with positive SST anomalies along the U.S. east coast and across the Gulf of Mexico like March 2012 (Figure 4c). Decay phases (Figure 12c) appear to be associated with negative SST anomalies across the eastern and central equatorial Pacific, but the results were not found to be statistically significant.

The findings for the warm LIS events were found to be sensitive to the threshold used to construct the composites. For example, if we only considered the LIS warm events whose intensities were greater than or equal to the 90[th] percentile of LIS warm event intensities, then all phases of LIS warm events would resemble the pattern shown in Figure 12b. In general, the positive SST anomalies across the eastern Pacific were found to become more intense as the percentile used to establish the threshold was increased from 50 to 90. Despite the lack of statistical significance in the composite plots, statistically relationships with SST anomalies were found when correlating DJF LIS temperature anomalies with DJF SST anomalies (Figure 3c). The identified correlation pattern was found to resemble the pattern shown in Figures 10b and 11b.

The SST composite analyses were also conducted using the dipole events for the 1979-2013, 1950-2013, and 1870-2013 periods. The resulting SST patterns were found to be like those shown in Figures 10, 11, and 12, which is not surprising given the strong correlation between the dipole index and LIS temperature anomalies. Thus, intense long-lived dipole patterns seem to have a tropical origin, suggesting that a key to better understanding LIS temperature events rests in a firmer understanding of tropical processes.

**4.4 Decadal Variability**

The results of the composite analyses suggest that dipole events may be associated with tropical SST patterns, but the time scale at which the SST patterns are most strongly associated with the dipole pattern cannot be inferred from the analysis. Thus, a wavelet coherence analysis was conducted to determine if the SST modes fluctuate coherently with the dipole pattern at a preferred time scale. The wavelet squared coherence was computed between the dipole index and indices for Niño 3 and Niño 4 metrics, but the results using the Niño 4 index were found to be most robust. As such, the results for the Niño 4 index analysis are only shown.

The results shown in Figure 13a indicate that the dipole and Niño 4 indices fluctuate coherently in the 64- to 256-month period band after 1930. The results suggest that stronger decadal-scale fluctuations in central equatorial Pacific SSTs are associated with larger decadal fluctuations in the dipole pattern. Given that the decadal-scale fluctuations in the dipole pattern contribute to the overall variance of the dipole index around 2012, the decadal-scale fluctuations must contribute to some extent to the intense dipole event of 2012. The results from the coherence analysis thus suggest that central equatorial Pacific SST fluctuations may have contributed to that intense dipole event.

The strong correlation between the EP/NP and dipole indices (Table 3) suggests that the coherence between the EP/NP and Niño 4 indices is also strong. The strong coherence was confirmed by computing the wavelet squared coherence between the EP/NP and Niño 4 indices for the 1950-2013 period. To perform the analysis, the missing values for the EP/NP index in December were filled by establishing a linear relationship between the EP/NP and dipole indices for all months but December. The linear relationship was obtained using a least-squares fit of a line, and it was used to fill missing EP/NP values based on the available December dipole index values.

As shown in Figure 14, the EP/NP index does, indeed, fluctuate coherently with the Niño 4 index. The coherence appears to be strong, and the global coherence spectrum shows arcwise significant global wavelet coherence in the 64-256 month period band. As shown by Schulte et al. (2018), the EP/NP pattern fluctuates strongly on quasi-decadal time scales, but no possible source of the variability was identified. We show in Figure 14 that the EP/NP variability on quasi-decadal time scales may be related to quasi-decadal fluctuations in central equatorial Pacific SSTs. Because the EP/NP pattern fluctuates coherently

with LIS water temperature on decadal time scales (Schulte et al. 2018), LIS water temperature should also fluctuate coherently with the Niño 4 index, though the LIS temperature time series is too short to draw strong conclusions.

**5 Conclusion**

This paper revealed that LIS events are associated with modes of tropical Pacific and North Pacific SST variability. The phases of the LIS events were found to depend on the spatial characteristics of the SST patterns. The onset of LIS cold events was shown to be associated with central equatorial Pacific SST anomalies, whereas the decay phase of such events was shown to coincide with canonical ENSO events. These results suggest that central Pacific El Niño events can be used to construct outlooks for the
onset of major LIS cold events. Similarly, information regarding the formation of canonical ENSO events could prove useful as guidance for assessing how likely a LIS cold event will end. Conversely, major LIS cold events could be used to anticipate the formation of El Niño events.

  The strong relationships identified between the dipole index and LIS temperature anomalies suggest that the dipole index should be incorporated into LIS temperature outlooks and possibly
temperature outlooks for other regions of the U.S. as well. The forecast skill associated with such outlooks will depend on the ability to predict the phase and intensity of the dipole pattern. The association between tropical SST patterns and the dipole pattern could prove useful in extended dipole pattern outlooks, contrasting with the AO index whose predictability is limited (Jung et al., 2011). The coherence between the Niño 4 index and indices for the EP/NP and dipole patterns supports the idea that extended dipole
pattern outlooks based on tropical SST patterns may be possible. More research, however, is needed to quantify the ability of dynamical weather and seasonal forecasting models to predict the pattern.

  Although not the focus of this paper, the dipole pattern may be an important temperature indicator for other estuaries across the Northeast U.S. The correlation pattern shown in Figure 9 suggests that the dipole pattern could be an important temperature indicator for the Delaware Bay and Chesapeake Bay
estuaries. The strong correlation between air temperature and the dipole index across Maine also suggests that the dipole pattern may contribute significantly to the variability of water temperature across the Gulf of Maine. The Gulf of Maine has experienced rapid warming during the past decade (Pershing et al.,

2015) and understanding the causes of the rapid warming has important implications for fisheries. Future work could therefore include understanding how the dipole pattern may have contributed to the rapid Gulf of Maine warming.

The results from the present analysis are consistent with temperature events that occurred after the study period considered in this study. For example, the cold period around February 2015 transitioned into a record warm period for most of the U.S. The record warm period lasting from September 2015 to December 2015 coincided with an extreme El Niño event (Hu and Federov, 2017). In agreement with our results, the February 2015 SST pattern strongly resembled the SST pattern shown in Figure 10b, which our results suggest occurs at the peak of LIS cold events. Furthermore, the SST pattern and extreme cold across the eastern U.S. occurred before the El Niño formation, which is also in agreement with the results from the present study. These recent events support the results from our study that indicate extended LIS temperature outlooks may be possible if information regarding ENSO flavors are incorporated into such outlooks.

**Data availability**

The 20th century reanalysis data is available at https://www.esrl.noaa.gov/psd/data/20thC_Rean/ and NCEP reanalysis data are available at https://www.esrl.noaa.gov/psd/data/gridded/data.ncep.reanalysis.html. The Hadley SST data are

5 available at https://www.metoffice.gov.uk/hadobs/hadisst/data/download.html. Monthly indices for the atmospheric climate modes can be found at https://www.esrl.noaa.gov/psd/data/climateindices/list/, while the long-term Nino 3.4 and Nino 4 indices are available at https://www.esrl.noaa.gov/psd/gcos_wgsp/Timeseries/. The Long Island Sound data is available at https://www.researchgate.net/publication/306256135_LIS_tmp_8113. The dipole index and forecasting

10 tools are available at http://justinschulte.com/forecasting/dipole.html.

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

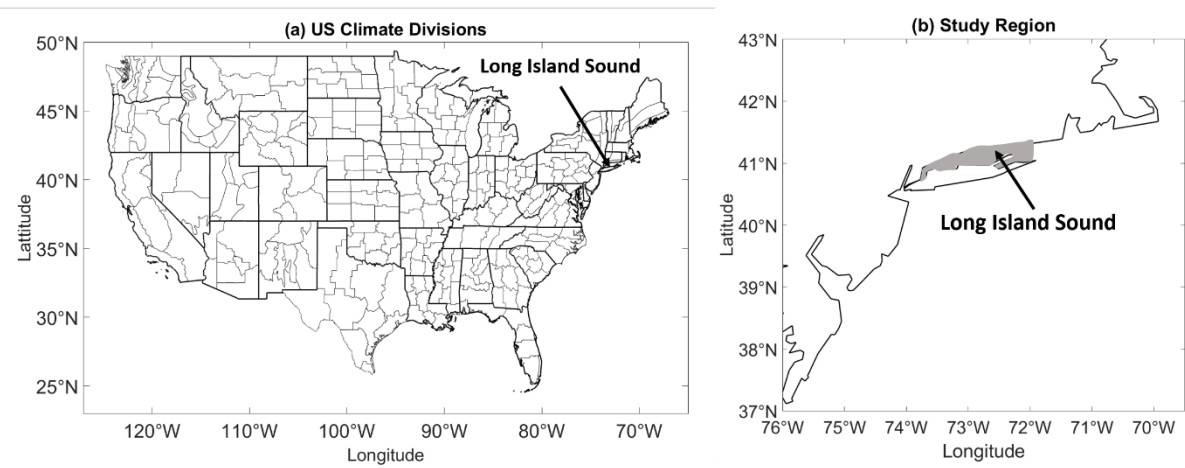

**Figure 1:** (a) 344 U.S climate divisions and (b) the LIS study region. Gray shading delineates the region used to calculate the LIS surface water temperature time series.

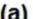

**Figure 2:** (a) The LIS surface temperature anomaly time series and (b) the corresponding event spectrum. Central Pacific El Nino events are indicated by CP and eastern Pacific El Nino Events are indicated by EP. Blue curves represent the 5 most intense negative LIS temperature events, while red curves represent the 5 most intense positive LIS temperature events. The length of the line segments in (b) represents the persistence of the LIS temperature events. The vertical axis corresponds to the intensity of the event.

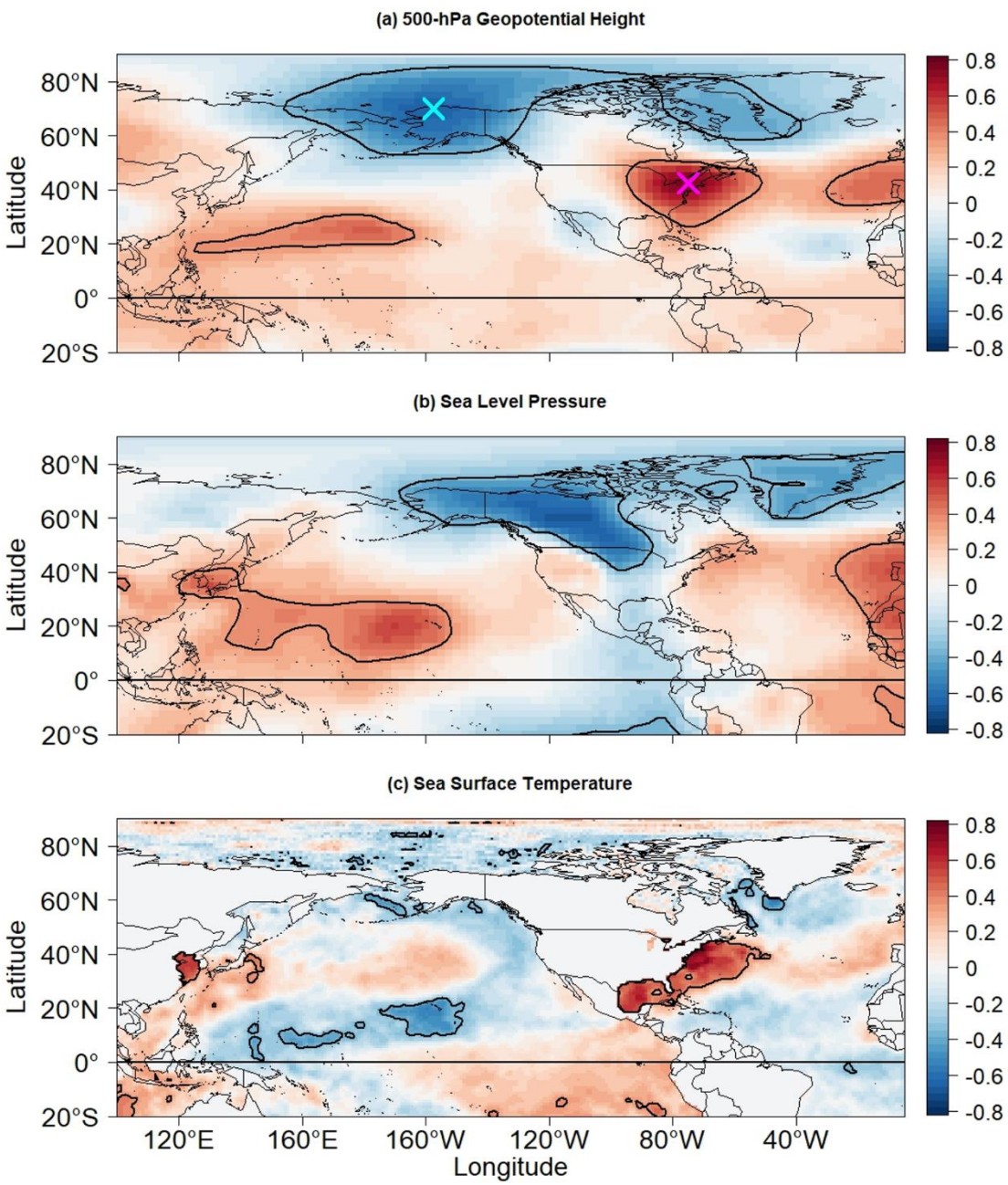

**Figure 3:** Correlation between LIS temperature anomalies and anomalies for (a) DJF 500-hPa geopotential height, (b) SLP, and (c) SST. Contours enclose regions of 5% statistical significance. Crosses in (a) mark the grid point locations used to construct the dipole index.

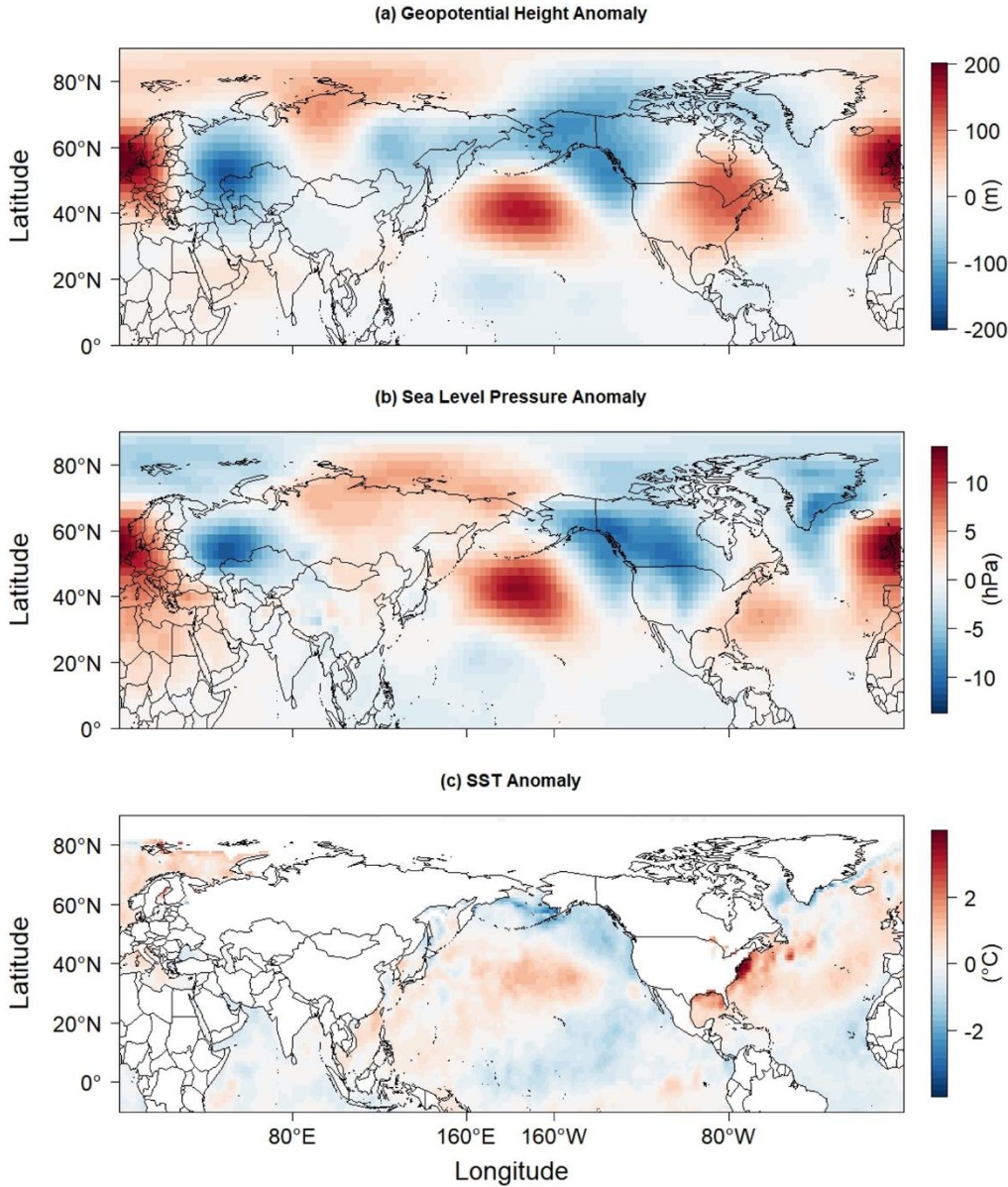

**Figure 4:** Anomalies for (a) 500-hPa geopotential height, (b) SLP, and (c) SST corresponding to the March 2012 LIS temperature event.

5    copP2logIN#

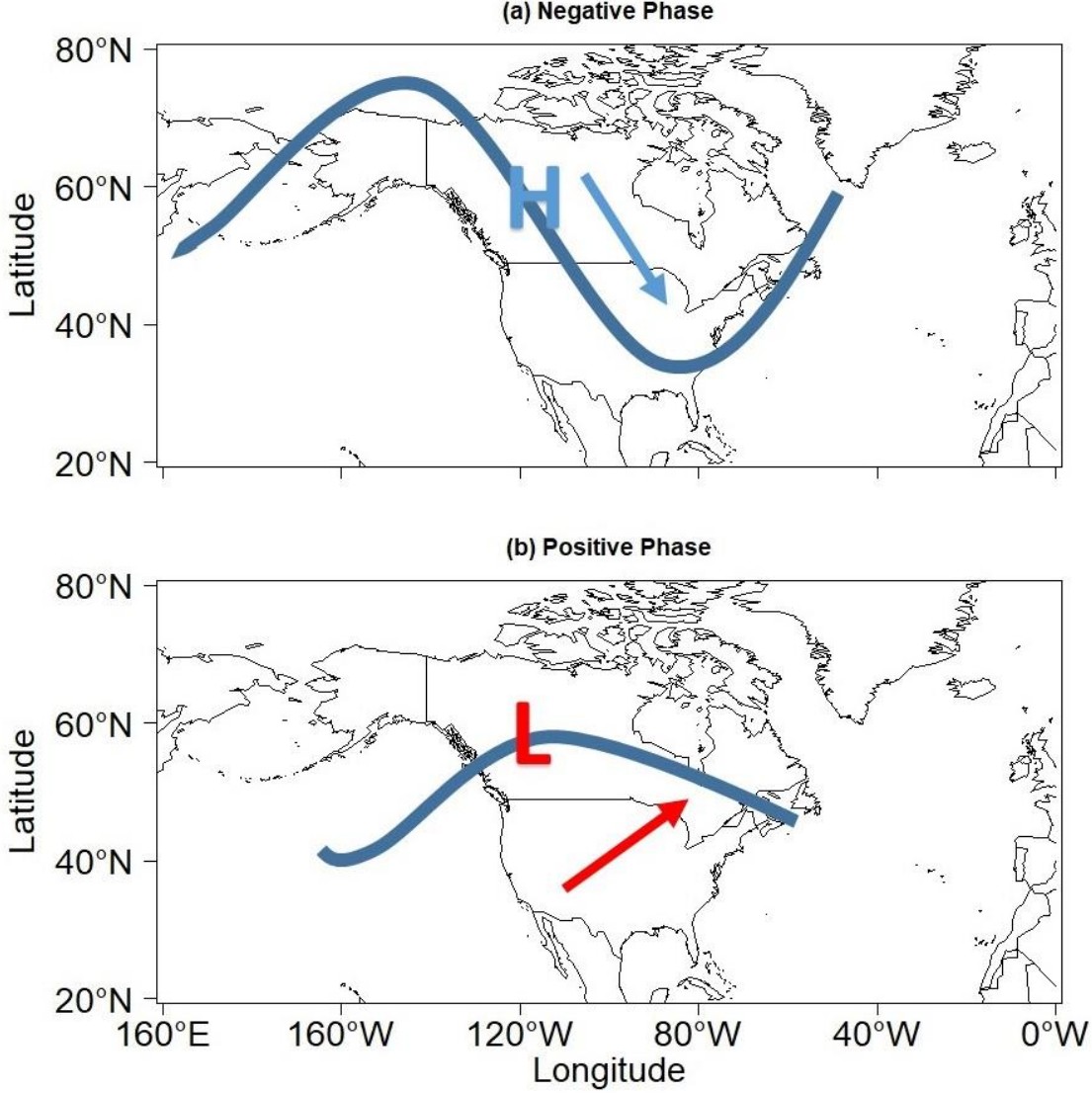

**Figure 5: (a)** Idealized schematic of atmospheric features occurring during negative LIS temperature events and negative
phases of the dipole pattern. (b) Same as (a) but for positive LIS temperature events and positive phases of the dipole pattern.
Thick blue curves represent the idealized Jetstream configuration, while the blue (red) arrow indicates the general movement
of cold (warm) air masses. High pressure is indicated with an H and low pressure is indicated with an L.

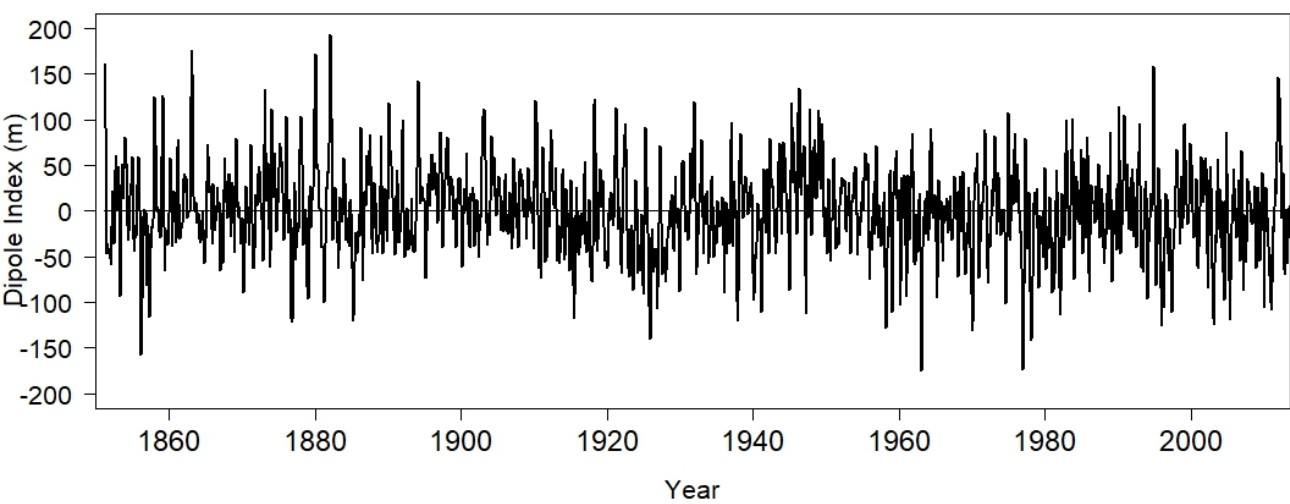

**Figure 6:** 3-month running mean of the dipole index.

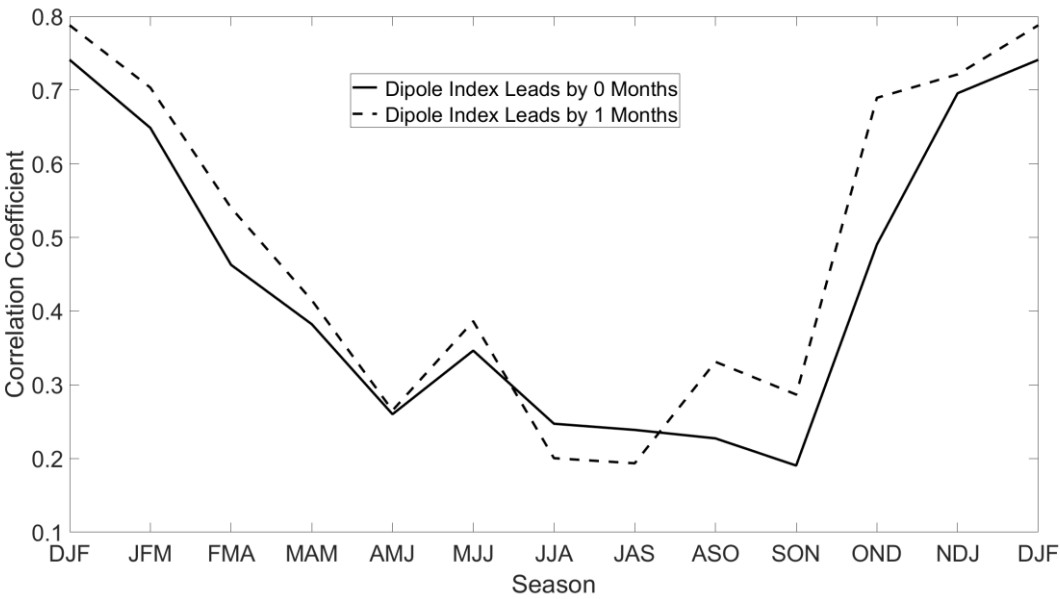

**Figure 7:** Lagged and simultaneous correlations between seasonally averaged LIS water temperature anomalies and the seasonally averaged dipole index. The dotted line represents the correlation between the dipole index of the prior season (dipole leads by 1 month) and water temperature anomalies for the season specified on the horizonal axis.

**Figure 8:** Correlation between 500-hPa geopotential height anomalies and indices for the (a) December dipole, (b) January dipole, and (c) January EP/NP patterns. Contours enclose regions of 5% statistical significance.

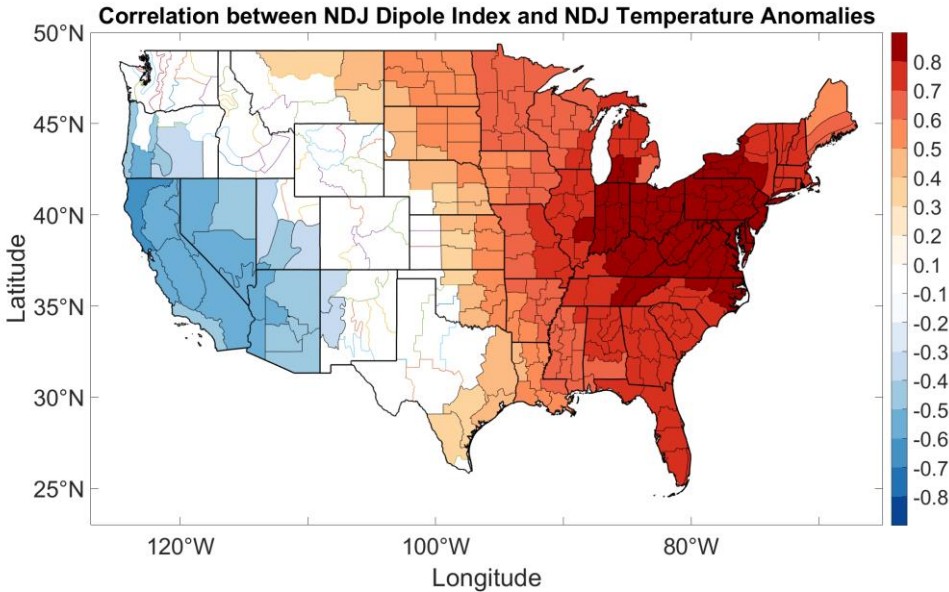

**Figure 9:** Correlation between the NDJ dipole index and NDJ temperature anomalies. Shaded climate divisions are those for which the corresponding correlation coefficients are statistically significant at the 5% significance level.

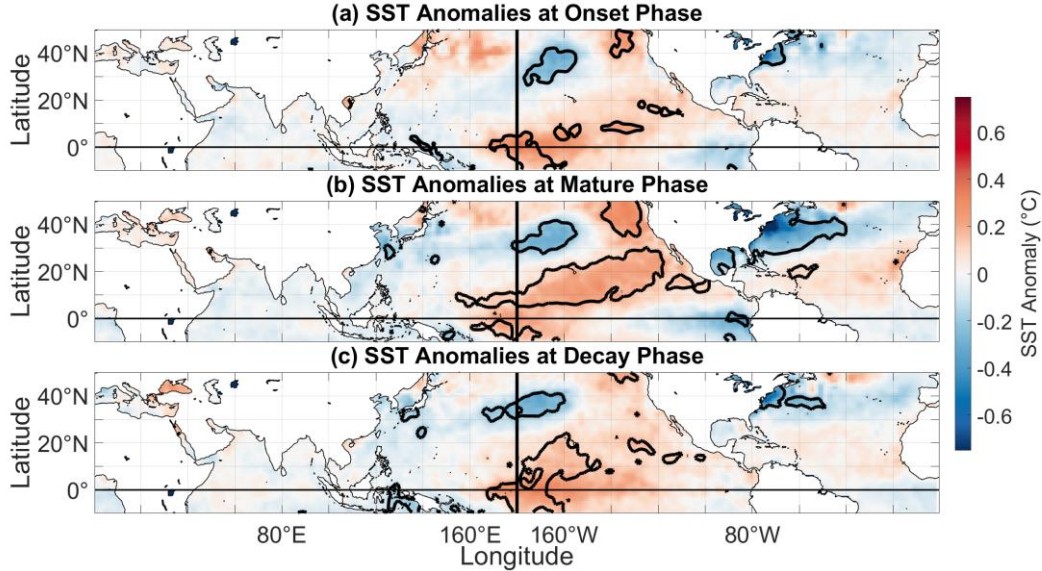

**Figure 10:** Composite mean SST anomalies corresponding to (a) onset, (b) mature, and (c) decay phases of negative LIS temperature events. Contours enclose regions of 5% statistical significance, as determined by a one sample *t*-test.

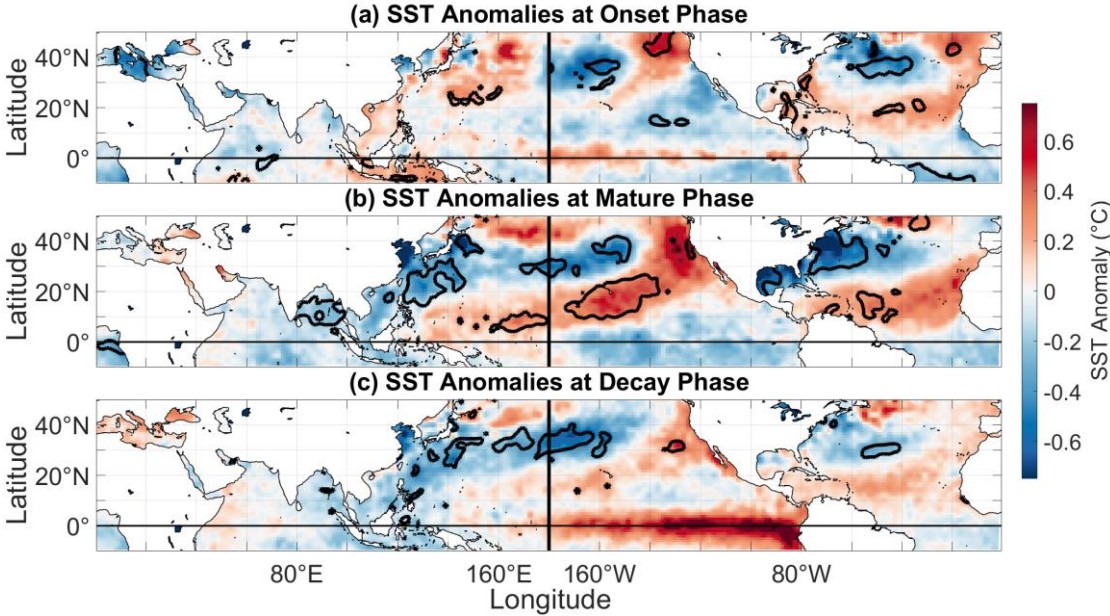

**Figure 11:** Same as Figure 10 but using the criterion that the LIS events fall below 1 one standard deviation from the mean intensity of all LIS cold events.

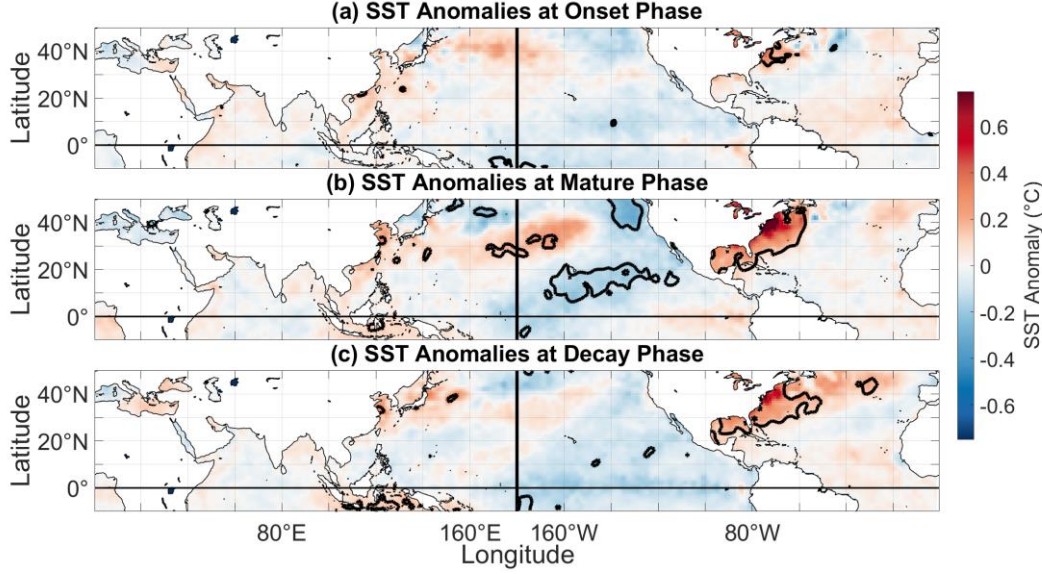

**Figure 12:** Same as Figure 10 except for positive LIS temperature events.

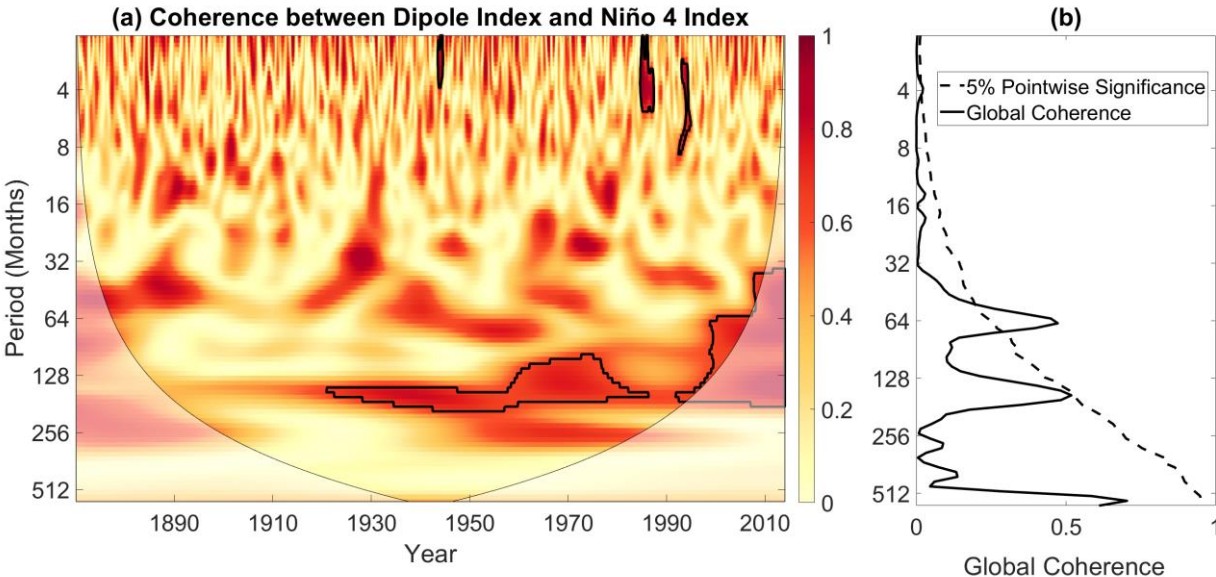

**Figure 13:** (a) Wavelet coherence between the dipole and Niño 4 indices. Contours enclose regions of 5% cumulative areawise significance. (b) The global coherence spectrum corresponding to (a). Dotted line is the 5% pointwise significance bound and the red curves indicate 5% arcwise significant coherence values.

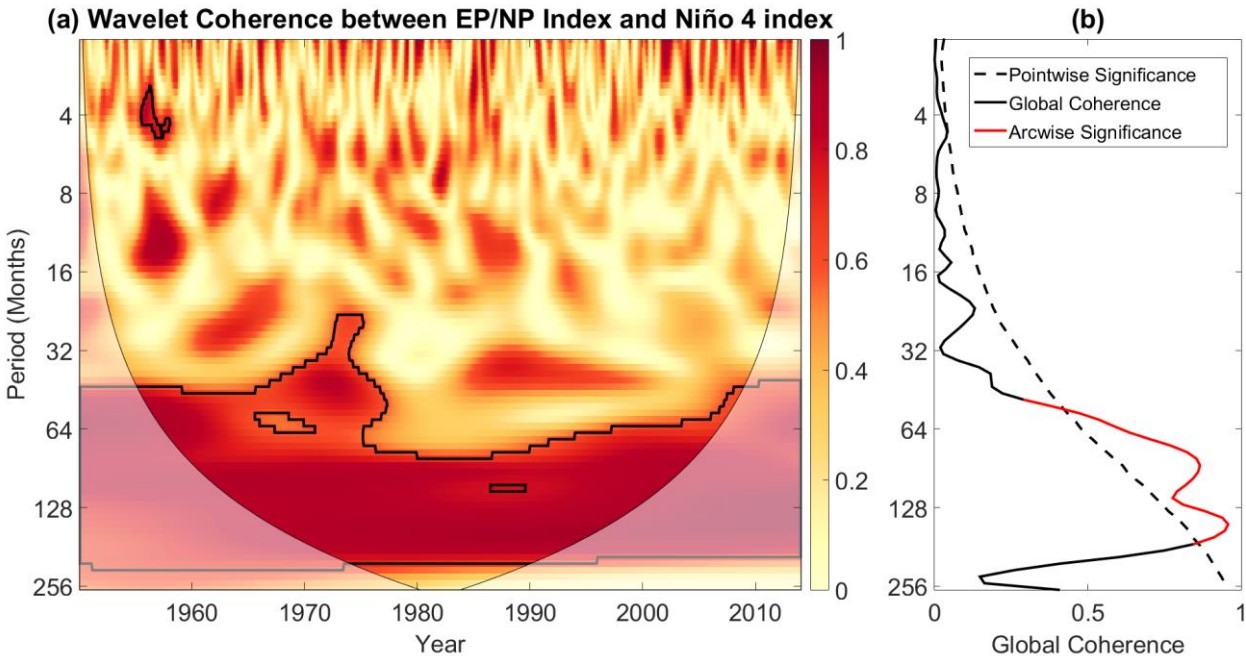

**Figure 14:** Same as Figure 13 but for the wavelet coherence between the EP/NP and Niño 4 indices.

Table 1. 10 LIS detrended temperature events ranked by the magnitude of their intensities.

| Intensity (°C) | Persistence (Months) | Peak | Onset | Decay |
|---|---|---|---|---|
| 20 | 13 | March 2012 | October 2011 | October 2012 |
| -13 | 14 | January 1982 | August 1981 | September 1982 |
| 13 | 15 | May 1991 | December 1990 | February 1992 |
| -13 | 12 | February 2003 | November 2002 | October 2003 |
| -12 | 14 | January 1996 | November 1995 | December 1996 |
| 11 | 6 | December 2001 | November 2001 | April 2002 |
| 10 | 7 | January 1983 | October 1982 | April 1983 |
| 9 | 14 | February 1998 | December 1998 | January 2000 |
| -8 | 8 | January 2011 | September 2010 | April 2011 |
| -8 | 5 | January 1981 | November 1980 | March 1981 |

Table 2. 10 dipole events in the 1851-2013 period ranked by the magnitude of their intensities. Results are based on the raw monthly dipole index time series.

| Intensity (m) | Persistence (Months) | Peak | Onset | Decay |
|---|---|---|---|---|
| 847 | 11 | February 1882 | May 1881 | March 1882 |
| 844 | 9 | January 1880 | October 1879 | June 1880 |
| 805 | 9 | March 2012 | July 2011 | March 2012 |
| -687 | 8 | January 1977 | July 1976 | February 1977 |
| -666 | 9 | January 2003 | October 2002 | June 2003 |
| -638 | 6 | January 1978 | December 1977 | May 1978 |
| -616 | 7 | September 1876 | July 1876 | January 1877 |
| -604 | 12 | August 1927 | May 1927 | April 1928 |
| 580 | 6 | January 1863 | December 1862 | May 1863 |
| 572 | 9 | December 1889 | November 1889 | July 1890 |

Table 3 Correlation between the dipole index and indices for 5 major climate modes of variability for the 1979-2013 period. Bold entries indicate 5% statistically significant correlation coefficients.

| | J | F | M | A | M | J | J | A | S | O | N | D |
|------|------|------|------|------|------|------|------|------|------|------|------|------|
| EPNP | **0.74** | **0.67** | **0.67** | **0.66** | **0.56** | **0.52** | **0.47** | **0.26** | **0.43** | **0.66** | **0.66** | ------- |
| WP | -0.28 | **-0.34** | -0.19 | **-0.37** | 0.21 | **0.34** | **0.42** | **0.33** | 0.00 | -0.31 | **-0.6** | **-0.56** |
| PNA | 0.29 | 0.0 | 0.0 | 0.0 | **0.35** | 0.12 | 0.12 | **0.46** | **0.43** | 0.30 | 0.15 | 0.0 |
| AO | **-0.6** | -0.18 | **-0.49** | -0.21 | **-0.52** | **-0.40** | **-0.45** | **-0.40** | **-0.37** | **-0.65** | **-0.58** | **-0.62** |
| NAO | **-0.59** | -0.15 | **-0.53** | -0.13 | **-0.50** | **-0.40** | -0.18 | -0.13 | -0.1 | **-0.55** | **-0.46** | **-0.59** |

Table 4. The mode number of the EOF pattern that most closely resembles the EP/NP pattern, the explained variance associated with the EOF pattern, and the correlation coefficient, $r$, computed between the corresponding principal component time series and the EP/NP index. The EOF pattern that most closely resembles the EP/NP pattern was determined by finding the EOF pattern whose principal component time series is most strongly correlated with the EP/NP index. The results are based on NCEP reanalysis for the 1979-2013 period.

| Quantity | J | F | M | A | M | J | J | A | S | O | N | D |
|---|---|---|---|---|---|---|---|---|---|---|---|---|
| r | 0.66 | 0.66 | 0.61 | 0.62 | 0.67 | 0.44 | 0.51 | 0.60 | 0.63 | 0.69 | 0.61 | ----- |
| Variance (%) | 14.4 | 5.5 | 3.2 | 5.0 | 7.0 | 1.9 | 4.3 | 5.2 | 4.7 | 6.0 | 4.4 | ----- |
| EOF Number | 2.0 | 6.0 | 7.0 | 6.0 | 4.0 | 11.0 | 5.0 | 5.0 | 6.0 | 5.0 | 7.0 | ----- |

**Table 5** El Nino events partitioned into Central (CP) and Eastern (EP) pacific types based on the categorization method of Yu et al. (2012). El Nino Events are defined based on the DJFM season. Third column provides the corresponding DJFM LIS temperature anomaly. A more complete table of El Nino events can be found in Johnson and Kosoka (2016).

| El Niño years | Event Type | LIS Temperature Anomaly (°C) |
|---|---|---|
| 1982-1983 | EP | 1.9 |
| 1986-1987 | EP | 0.2 |
| 1987-1988 | CP | -0.2 |
| 1991-1992 | CP | 0.3 |
| 1994-1995 | CP | 0.7 |
| 1997-1998 | EP | 0.9 |
| 2002-2003 | CP | -1.5 |
| 2004-2005 | CP | -0.4 |
| 2006-2007 | EP | 0.7 |
| 2009-2010 | CP | -0.6 |

