# Peer review of "Long Island Sound Temperature Variability and its Associations with the Ridge-trough Dipole and Tropical Modes of Sea Surface Temperature Variability"

_Ocean Science, 2018_

## Referee Comment (RC1) · Anonymous Referee #1 · 23 Apr 2018

**Review of**

"Long Island Sound Temperature Variability and its Associations with the Ridge-trough Dipole and Tropical Modes of Sea Surface Temperature Variability"

by Justin A. Schulte and Sukyoung Lee

**General comments:**

The present study examines the variability of surface water temperatures at Long Island Sound (LIS) and how cold and warm events in this region are related to a dipole of atmospheric ridge and trough in the North Pacific, and to sea surface temperatures (SST) in the tropical Pacific Ocean.

The article addresses a relevant topic with important implications for the climate forecast and impacts community and overall appears to be technically sound. It makes interesting use of quite novel methods (e.g. statistical significance testing of event decomposition or wavelet analyses) to examine the variability and potential predictability of the LIS thermal system. However, the authors do not discuss any possible physical mechanism that could be responsible for these statistical associations (as stated in the Abstract). The choice of season for each analysis is not always clear and well justified which makes it quite difficult for the reader to understand the (lagged) relationship between LIS water temperatures, the atmospheric dipole and tropical SST variability. I think the overall structure of the manuscript could be improved, by being more succinct, by being careful not to over interpret results but rather highlighting how these results are new compared to the previous literature.

**Specific comments:**

Abstract:

The physical mechanisms are not (or poorly) discussed in the text. The 2012 ocean heat wave across the mid-Atlantic Bight to my knowledge has not been discussed in detail either.

1. Introduction:

It would be useful to explain why the LIS is an important region to study (in terms of impacts), and perhaps to describe in more detail previous results on the importance of the EP/NP pattern.
If possible, please also add a reference discussing the lack of relationship with the Gulf Stream and NAO.

3. Methods:

Page 5 line 7: How many adjacent points are required to be considered an event?

4. Results:

Fig. 2: Where are the vertical dotted lines representing the occurrence of canonical and East Pacific El Nino? Perhaps it would be useful to highlight the most intense warm and cold LIS events in red and blue (just a suggestion).

Section 4.1:
It would be nice to clearly explain the added value of the event spectrum compared to the simple time series. The paragraph could also be shortened here.
In what seasons are these extreme temperature anomalies more likely to occur? How long do they last? More discussion of Table 1 would be helpful here.

Section 4.2:
Fig 3: How different do patterns look in each season? How coherent is this atmospheric dipole on different vertical levels? It would be worth detailing the possible physical mechanisms operating behind this forcing from the atmospheric dipole onto LIS water temperature variability.
Fig. 4: I wonder how useful this figure is. It is very noisy and difficult to discern the 2012 event.

Section 4.3:
The composite analyses are interesting (particularly the discussion of specific LIS events and El Nino years), with potentially important implications for the forecasting of LIS temperatures. However it is not clear how many events compose each of the LIS composites (counted in months, seasons?), which makes it difficult to really interpret in the context of ENSO and predictability.
Perhaps it would be useful to show the correlation between the dipole index and SST anomalies (page 12, line 21). It might also be important to discuss the overall added value of the dipole index compared to the EP/NP index used in previous work.

Finally, while the references to extreme LIS events are insightful, I think it might be helpful to consider the 2012 warm event more clearly as a case study throughout the text. It would help with the readability of the text and our understanding of the climate teleconnections mentioned.

---

## Referee Comment (RC2) · Anonymous Referee #2 · 9 Aug 2018

In this paper, the authors performed a statistical analysis to show that Long Island Sound (LIS) surface temperature is linked to mid-tropospheric geopotential height difference between the US east coast and Alaska, which is referred to as ridge-trough dipole in this study. By performing composite analysis, they further argue that the dipole is forced by sea surface temperature anomalies in the central equatorial Pacific.

From Figure 2, it is quite clear that LIS surface temperature is determined by the meridional shift of the atmospheric jet and the associated winter storm track across the US east coast. For instance, a northward shift of the storm track will warm up the subtropical North Atlantic and the mid-Atlantic bight, and cool the subpolar North Atlantic. The associated increase in the subtropical high will further increase the trade wind cooling the tropical North Atlantic. This is a well-known process, also known as North Atlantic tripole SST mode, which can be well observed in Figures 7 and 8. The North Atlantic tripole SST mode can be triggered by NAO, ENSO teleconnection and PNA, which is also very well known (e.g., Deser & Blackmon, 1993). Therefore, it is difficult for me to justify the publication of this manuscript in Ocean Science. I also feel that it is not well justified in the introduction why we need to study LIS surface temperature variability.

Additionally, I am not convinced that the geopotential anomaly over Alaska is anything to do with LIS surface temperature. The atmospheric circulation anomalies linked to LIS surface temperature anomalies shown in Figure 3 does not appear to be one of the leading modes of NH atmospheric variability (Wallace & Gutzler, 1981) or typical ENSO teleconnection patterns. The East Pacific - North Pacific (EP- NP) pattern, which is inactive in boreal winter, does not look like Figure 3 either. I am also confused why the authors use a numerical model data instead of observations for the LIS surface temperature. And, it is not clear what depth is at "the first vertical level". There are also many statements that do not match with the corresponding figures. I am sorry that I cannot be more positive.

---

## Author Comment (AC1) · 6 Sep 2018

General comments: The present study examines the variability of surface water temperatures at Long Island Sound (LIS) and how cold and warm events in this region are related to a dipole of atmospheric ridge and trough in the North Pacific, and to sea surface temperatures (SST) in the tropical Pacific Ocean.

**The article addresses a relevant topic with important implications for the climate forecast and impacts community and overall appears to be technically sound. It makes interesting use of quite novel methods (e.g. statistical significance testing of event decomposition or wavelet analyses) to examine the variability and potential predictability of the LIS thermal system. However, the authors do not discuss any possible physical mechanism that could be responsible for these statistical associations (as stated in the Abstract). The choice of season for each analysis is not always clear and well justified which makes it quite difficult for the reader to understand the (lagged) relationship between LIS water temperatures, the atmospheric dipole and tropical SST variability. I think the overall structure of the manuscript could be improved, by being more succinct, by being careful not to over interpret results but rather highlighting how these results are new compared to the previous literature.**

The authors the appreciate the comments provided and many changes have been to the original manuscript. Our responses to the comments are in plain text and the original comments are in bold text (bold text). Changes to the manuscript include the addition of a March 2012 case study, the inclusion of motivation for constructing the dipole index, and the addition of text describing the physical mechanism behind the dipole relationship with Long Island Sound temperature variability. Specific changes are described below.

 Specific comments:

 Abstract:

 **The physical mechanisms are not (or poorly) discussed in the text. The 2012 ocean heat wave across the mid-Atlantic Bight to my knowledge has not been discussed in detail either.**

The authors agree that a physical mechanism is needed to explain the relationships identified in this study. To address the issue, we have included a discussion about

how our dipole pattern is linked to LIS temperature variability using well-known ideas in meteorology.

**1. Introduction:**

**It would be useful to explain why the LIS is an important region to study (in terms of impacts), and perhaps to describe in more detail previous results on the importance of the EP/NP pattern. If possible, please also add a reference discussing the lack of relationship with the Gulf Stream and NAO.**

The authors agree that the reason for studying the LIS needs to be better motivated. As such, a discussion describing the relevance of the LIS to fisheries was included in the revised manuscript. The authors also agree more background material is needed. We now highlight previous work examining EP/NP pattern impacts and use those studies to motivate the construction of our dipole index. A reference to the Gulf Stream and NAO linkages with LIS temperature has also been included.

**3. Methods:**

**Page 5 line 7: How many adjacent points are required to be considered an event?**

According to the definition of an event, there is no number of adjacent points required for an event. This lack of length restriction is now made more explicit in the methods section for clarity.

**4. Results:**

**Fig. 2: Where are the vertical dotted lines representing the occurrence of canonical and East Pacific El Nino? Perhaps it would be useful to highlight the most intense warm and cold LIS events in red and blue (just a suggestion).**

The authors appreciate the suggestion regarding the clarity of the figure. The most intense LIS events are now highlighted with color; the CP and EP EL Ninos are now indicated with acronyms in the figure.

**Section 4.1: It would be nice to clearly explain the added value of the event spectrum compared to the simple time series. The paragraph could also be shortened here. In what seasons are these extreme temperature anomalies more**

**likely to occur? How long do they last? More discussion of Table 1 would be helpful here.**

 The authors agree that the added value of the event spectrum has not been clearly explained. One reason for choosing the event spectrum approach is that we can treat time periods in which LIS temperature anomalies are of similar sign as individual events, which helps account for autocorrelation in the time series. Accounting for autocorrelation makes our results more statistically robust. Also, by separately looking at negative and positive events we are better able to uncover differences in the intensity of negative and positive events. The event spectrum also provides an easy way to objectively calculate the persistence of events, contrasting with lag-1 autocorrelation coefficients that need to be calculated over some predetermined time interval. Thus, persistence as measured using autocorrelation would be a function of the time window used.

We found that intense LIS temperature events tend to occur most frequently during the cool season (November-March), presumably because atmospheric forcing is stronger during the cool season. On average, LIS temperature events last about 2 months, but the most intense LIS temperature events can span several seasons. A discussion to this effect was incorporated into Section 4.1 of the revised manuscript to help better explain Table 1.

**Section 4.2:  Fig 3: How different do patterns look in each season? How coherent is this atmospheric dipole on different vertical levels? It would be worth detailing the possible physical mechanisms operating behind this forcing from the atmospheric dipole onto LIS water temperature variability. Fig. 4: I wonder how useful this figure is. It is very noisy and difficult to discern the 2012 event.**

The atmospheric pattern related to LIS temperature variability is generally the same for each season, but the relationships are strongest in the winter. We now note this seasonality in the revised manuscript. The authors note that the LIS is well-mixed (especially during the winter) so that the atmospheric pattern related to LIS water temperature at one vertical level is the same as those at other vertical levels. This is now mention in the text in Section 4.2. The authors agree that a physical mechanism behind the dipole pattern relationship with LIS temperature is needed. We now describe the physical mechanism in terms of the jet stream and surface pressure systems. For example, we now describe how the ridge over Alaska

induces a downstream anticyclone that can be inferred to advect Arctic air from northern portions of North America into the LIS region. The authors have deleted Figure 4 of the original manuscript but included a schematic of the physical mechanism underlying the LIS temperature-dipole relationship.

**Section 4.3: The composite analyses are interesting (particularly the discussion of specific LIS events and El Nino years), with potentially important implications for the forecasting of LIS temperatures. However it is not clear how many events compose each of the LIS composites (counted in months, seasons?), which makes it difficult to really interpret in the context of ENSO and predictability. Perhaps it would be useful to show the correlation between the dipole index and SST anomalies (page 12, line 21). It might also be important to discuss the overall added value of the dipole index compared to the EP/NP index used in previous work.**

The authors agree that more information regarding the number of events is needed. The authors note that the events are not counted in seasons or months. Rather, the events are an arbitrary length. However, many of the events used in the composites have associated peaks in the winter. Thus, the composites shown in the composite figures generally reflect winter conditions. This information is now provided in Section 4.3 to help guide the reader. The authors agree that it would be useful to show the correlation between the dipole index and SST because the correlation approach is more standard and will help readers interpret our results. We therefore have included a new figure in the manuscript for comparison.

We also now include a discussion highlighting the value of our dipole index. We now point out, for example, that the dipole index is optimized to explain the variance of LIS temperature, whereas the EP/NP index is constructed such that it explains a relatively large fraction of geopotential height variability. As such, the dipole index is a better predictor of LIS temperature than the EP/NP index. Also, the EP/NP index is curiously not defined in December, further complicating its use in a forecasting setting. We now mention that the dipole index is needed because patterns extracted from an EOF analysis are unable to capture LIS temperaure variability in December.

**Finally, while the references to extreme LIS events are insightful, I think it might be helpful to consider the 2012 warm event more clearly as a case study**

**throughout the text. It would help with the readability of the text and our understanding of the climate teleconnections mentioned.**

The authors agree that the inclusion of a case study will help readability of the manuscript. Thus, we included composite plots of the March 2012 event. We discuss in more detail the atmospheric and ocean features present during that event as well.

---

## Author Comment (AC2) · 6 Sep 2018

**In this paper, the authors performed a statistical analysis to show that Long Island Sound (LIS) surface temperature is linked to mid-tropospheric geopotential height difference between the US east coast and Alaska, which is referred to as ridge-trough dipole in this study. By performing composite analysis, they further argue that the dipole is forced by sea surface temperature anomalies in the central equatorial Pacific. From Figure 2, it is quite clear that LIS surface temperature is determined by the meridional shift of the atmospheric jet and the associated winter storm track across the US east coast. For instance, a northward shift of the storm track will warm up the subtropical North Atlantic and the mid-Atlantic bight, and cool the subpolar North Atlantic. The associated increase in the subtropical high will further increase the trade wind cooling the tropical North Atlantic. This is a well-known process, also known as North Atlantic tripole SST mode, which can be well observed in Figures 7 and 8. The North Atlantic tripole SST mode can be triggered by NAO, ENSO teleconnection and PNA, which is also very well known (e.g., Deser & Blackmon, 1993). Therefore, it is difficult for me to justify the publication of this manuscript in Ocean Science.**

The authors the appreciate the comments provided and many changes have been to the original manuscript. Our responses to the comments are in plain text and the original comments are in bold text (bold text). Changes to the manuscript include the addition of a March 2012 case study, the inclusion of motivation for constructing the dipole index, and the addition of text describing the physical mechanism behind the dipole relationship with Long Island Sound temperature variability. Specific changes are described below.

Although the authors do see some evidence for the tripole mode in Figures 7 and 8, the signal is not particularly robust because the composite means are mainly statistically significant along the east coast of the United States. The lack of relationship with the tripole mode is consistent with how LIS temperature anomalies are not strongly linked to the NAO and PNA patterns that have been identified in previous work as excitation mechanisms for the tripole mode. That lack of relationship between LIS temperature anomalies and the tripole SST mode were confirmed by correlating LIS temperature anomalies with SSTs across the North Atlantic. It was found that LIS temperature anomalies are only significantly correlated with Atlantic SSTs along the east coast US and not across other regions where the tripole mode centers are located. Thus, the LIS mode is new and largely different from the tripole mode identified in previous work.

**I also feel that it is not well justified in the introduction why we need to study LIS surface temperature variability. Additionally, I am not convinced that the geopotential anomaly over Alaska is anything to do with LIS surface temperature.**

The authors agree that the physical mechanism behind the dipole pattern-LIS temperature relationship is not well-described in the original manuscript. As such, in the revised manuscript, a physical mechanism behind the connection between the Alaskan height anomaly and LIS temperature anomalies is now described. We have also included a schematic. The importance of the geopotential height anomaly over Alaska is now described using fundamental principles in meteorology. During a negative phase of the dipole pattern (ridge over Alaska and trough over the eastern US), the ridge over Alaska supports a surface anticyclone just upstream from the upper-level trough over the eastern US. The surface anticyclonic flow then is responsible for the advection of cold continental air across the Northeast US region, amplifying the trough over the eastern US.

**The atmospheric circulation anomalies linked to LIS surface temperature anomalies shown in Figure 3 does not appear to be one of the leading modes of NH atmospheric variability (Wallace & Gutzler, 1981) or typical ENSO teleconnection patterns. The East Pacific - North Pacific (EP- NP) pattern, which is inactive in boreal winter, does not look like Figure 3 either.**

While the authors agree that the dipole pattern does not resemble many of the well-known leading modes of variability (e.g NAO, PNA, and WP patterns), the dipole pattern is very important to LIS temperature variability. EOF patterns are modes that are extracted from an analysis that seeks patterns that explain the most variance within a data set. These patterns need not be relevant to climate variability across a given region. In fact, Schulte et al. (2016), Schulte et al. (2017a,b) and Schulte et al. (2018) found that the common leading modes do not explain much of the temperature and precipitation variability across the Northeast US. Thus, new patterns most be identified, and corresponding indices need to be constructed that can explain the temperature and precipitation variability. One alternative approach to EOF analysis is the continuum approach (Frankze and Feldstein, 2005; Johnson et al. 2008) in which atmospheric patterns are viewed as falling on a continuum, contrasting with EOF analysis that statistically partitions the atmosphere into a discrete finite set of patterns. Each location of the globe has a teleconnection pattern associated with it and the pattern may or may not resemble the leading modes extracted from an EOF analysis, much like how an arbitrary vector may not resemble any of the standard Cartesian basis vectors. To find a relevant teleconnection in the continuum, we adopted an one point correlation map approach, which resulted in our dipole pattern explaining 64% LIS temperature variability. This pattern is very similar to the EP/NP pattern, which we found to be a leading mode of variability, as determined by an EOF analysis (Table 4). In the introduction section of the revised manuscript, we motivate our construction of the dipole pattern by explaining how EOF patterns cannot capture precipitation and temperature variability across the Northeast US very well. We then discuss the added value of using the dipole index in the results section.

The authors disagree with the idea that the EP/NP is inactive in winter. The January and February EP/NP pattern very closely look like our dipole pattern for the same months, as can be confirmed by correlating indices for the EP/NP and dipole patterns during those months (Table 3). In fact, we found that correlation between the dipole index and the EP/NP index is strongest during the months of November, January, and February. These correlations are shown in Table 3 of the original manuscript and confirm that the EP/NP resembles the dipole pattern. Moreover, the relationship strength between the EP/NP and dipole indices increases from summer to winter, suggesting that the December dipole pattern is the EP/NP pattern and that the EOF analysis is unable to capture the EP/NP pattern in December. In fact, a comparison of the November and January EP/NP patterns with the December dipole pattern reveals a strong similarity among the patterns, providing more evidence that the EP/NP pattern is active in winter and that the dipole pattern is an important driver of LIS temperature variability. In the revised manuscript, we use these findings to motivative the construction of our dipole index that can be unambiguously defined for all calendar months.

**I am also confused why the authors use a numerical model data instead of observations for the LIS surface temperature. And, it is not clear what depth is at "the first vertical level".**

The authors agree that it is important to mention why numerical model data are used. For the Long Island Sound, observations are temporally and spatially sparse so that the output from a numerical model is

better suited for the event spectrum analysis that requires continuous data. The data quality has been evaluated and shown to agree well with observations in previous work (Georgas et al., 2016).

**There are also many statements that do not match with the corresponding figures.**

In the revised manuscript, inconsistences between the text and figures have been remedied throughout.

Franzke, C., and S. B. Feldstein, 2005: The continuum and dynamics of Northern Hemisphere teleconnection patterns. J. Atmos. Sci., 62, 3250–3267.

Georgas, N., Yin, L., Jiang, Y., Wang, Y., Howell, P., Saba, V., Schulte, J., Orton, P., and Wen, B.: An open-access, multi decadal, three-dimensional, hydrodynamic hindcast dataset for the Long Island Sound and New York/New Jersey Harbor Estuaries., J. Mar. Sci. Eng., 4, 48, 2016.

Johnson, N. C., S. B. Feldstein, and B. Tremblay, 2008: The continuum of Northern Hemisphere teleconnection patterns and a description of the NAO shift with the use of self-organizing maps. J. Climate, 21, 6354–6371.

Schulte, J. A., Georgas, N., Saba, V., Howell, P.: North Pacific Influences on Long Island Sound Temperature Variability, J. Clim., https://doi.org/10.1175/JCLI-D-17-0135.1, 2018.

Schulte, J. A., Najjar, R.G, and Li, M.: Impacts of Climate Modes on Streamflow in the Mid-Atlantic Region of the United States, J. Hydrology: Regional Studies, 5, 80-99, 2016.

Schulte, J.A.; Georgas, N.; Saba, V.; Howell, P. Meteorological Aspects of the Eastern North American Pattern with Impacts on Long Island Sound Salinity. J. Mar. Sci. Eng. 2017b, 5, 26.

Schulte, J. A. , Najjar, R.G, Lee, S.: Salinity and Streamflow Variability in the Mid-Atlantic Region of the United States and its Relationship with Large-scale Atmospheric Circulation Patterns, Journal of Hydrology, 550, 65-79. 2017a.